# TDLAS Monitoring of Carbon Dioxide with Temperature Compensation in Power Plant Exhausts

**Xiaorui Zhu [1,3], Shunchun Yao [1,3,*] , Wei Ren [2] , Zhimin Lu [1,3] and Zhenghui Li [1,3]**

[1] School of Electric Power, South China University of Technology, Guangzhou 510640, China; 201620110487@mail.scut.edu.cn (X.Z.); zhmlu@scut.edu.cn (Z.L.); epzhenghui@mail.scut.edu.cn (Z.L.)

[2] Department of Mechanical and Automation Engineering, The Chinese University of Hong Kong, Shatin, Hong Kong SAR 2912, China; renwei@mae.cuhk.edu.hk

[3] Guangdong Province Engineering Research Center of High Efficiency and Low Pollution Energy Conversion, Guangzhou 501640, China

*   Correspondence: epscyao@scut.edu.cn; Tel.: +86-139-2515-0807



**Featured Application: Applying to complex industrial environment, TDLAS monitoring with temperature compensation can come handy.**

**Abstract:** Temperature variations of flue gas have an effect on carbon dioxide ($CO_2$) emissions monitoring. This paper demonstrates accurate $CO_2$ concentration measurement using tunable diode laser absorption spectroscopy (TDLAS) with temperature compensation methods. A distributed feedback diode laser at 1579 nm was chosen as the laser source for $CO_2$ measurements. A modeled flue gas was made referring to $CO_2$ concentrations of 10–20% and temperatures of 298–338 K in the exhaust of a power plant. Two temperature compensation methods based on direct absorption (DA) and wavelength modulation (WMS) are presented to improve the accuracy of the concentration measurement. The relative standard deviations of DA and WMS measurements of concentration were reduced from 0.84% and 0.35% to 0.42% and 0.31%, respectively. Our experimental results have validated the rationality of temperature compensations and can be further applied for high-precision measurement of gas concentrations in industrial emission monitoring.

**Keywords:** carbon dioxide monitoring; absorption spectroscopy; temperature compensation; wavelength modulation

## 1. Introduction

Carbon dioxide ($CO_2$) is a major component of greenhouse gases. Measurements of $CO_2$ in large emission sources such as coal-fired power plants provide an effective support of carbon emission statistics. Outlet ducts in power plants provide a challenging detection environment for gas absorption detection due to the instability and poor timeliness. Compared with the current spectroscopic techniques such as Fourier transform infrared (FTIR) [1] and non-dispersive infrared (NDIR) spectroscopy [2], tunable diode laser absorption spectroscopy (TDLAS) with high sensitivity and short response time [3] offers significant advantages for in situ $CO_2$ measurements in the hot, humid, and dusty environments of power plants. With the rapid advancements in semiconductor laser industry, TDLAS has been widely applied in different fields such as combustion diagnostics [4–8], isotopic analysis [9–13], and atmospheric gas detection [14–18]. Because of their exquisite constituents and compact packaging, as well as low cost, tunable semiconductor diode lasers are nearly the ideal source for highly-sensitive and highly-spectral-resolved measurements in industrial fields [19]. However, Doppler-broadening at high temperatures alters a larger portion of the integrated absorbance

for a particular transition to the far-wing region. For in-situ $CO_2$ measurements in outlet ducts, the temperature-varied absorption line intensity leads to measurement error in both direct absorption (DA) and wavelength modulation spectroscopy (WMS) signals.

Numerous studies have been performed to investigate the influence of temperature variation on TDLAS measurement. Many of them used spectroscopic parameters such as line intensity [20] from databases like HITRAN (High Resolution Transmission) or HITEMP (High Temperature Molecular Spectroscopic). Nevertheless, these databases, which mainly derive from theoretical calculation, may have some uncertainty while applied in high-temperature measurement. Several of previous studies [21–25] measured spectral parameters such as line intensity, line position, and broadening coefficients in a harsh experimental condition. Under a certain range of pressure and temperature, they employed these spectroscopic parameters for temperature compensation. Empirical corrections to the temperature dependence of the absorption for $CO_2$–$CO_2$, $CO_2$–$N_2$, and $CO_2$–$O_2$ at 193–300 K and the wavelength near 4.3 μm are discussed [26,27]. These corrections are applied via the corrective shape χ-function to the line shape function of individual absorption features. Perrin et al. [28] continued the studies at 4.3 μm to develop a temperature dependent χ-function for corrections of $CO_2$–$CO_2$ and $CO_2$–$N_2$ collisions at temperatures up to 800 K. Additionally, Zhang et al. [29] proposed a 1f ratio method to correct the harmonic signals affected by temperature fluctuations. They employed a DFB (Distributed Feedback) laser at 760.77 nm to monitor 21% oxygen in the range of 300–900 K and the results showed that the proposed method was effective for minimizing the influence of temperature changes. Shu et al. [30] introduced a temperature correction coefficient for HCl concentration using TDLAS at 1.7 μm. Zhang et al. [31] studied the temperature influence on the detection of ammonia slip based on TDLAS. The empirical formula for temperature compensation was proposed and the error was reduced to about 5.1%. Qi et al. [32] simulated the process of direct TDLAS measurement of $H_2O$ based on the HITRAN spectra database and gave the temperature correction curve in atmosphere detection. A temperature correction function was optimized based on open-path TDLAS for determining ammonia emission rules in soil environments [33]. Liu et al. [34] combined regularization methods with TDLAS to measure non-uniform temperature distribution, relying on the measurements of 12 absorption transitions of water vapor from 1300 nm to 1350 nm. The results showed that regularization methods are less sensitive to the noise caused by temperature variation.

Despite some temperature compensation methods applied in industrial process monitoring, little work is found in practical emission measurements of $CO_2$ at 1579 nm in power plants. In this paper, TDLAS experiments were conducted in a temperature field simulating the outlet duct of a power plant. Both DA and WMS methods are used to measure $CO_2$ concentrations at different temperatures. Two temperature compensation methods are used to reduce the measurement errors.

## 2. Absorption Spectroscopy

### 2.1. Direct Absorption Spectroscopy

The fundamental theory that governs absorption spectroscopy for narrow-linewidth radiation sources is embodied in the Beer–Lambert law (Equation (1)). The ratio of the transmitted intensity $I_t$ and initial (reference) intensity $I_0$ of laser radiation through an absorbing medium at a particular frequency $v$ is exponentially related to the transition line intensity $S(T)$ ($cm^{-2} \cdot atm^{-1}$), line-shape function $\varphi(v)$ (cm), total pressure $P$ (atm), volume concentration of the absorbing species $X$, and the path length $L$ (cm):

$$\frac{I_t}{I_0} = \exp[-PS(T)\phi(v)XL] \tag{1}$$

If both sides of Equation (1) are computed logarithmically and integrated over the whole frequency domain, the volume concentration $X$ is described by [34]:

$$X = \frac{\int_{-\infty}^{+\infty} -\ln\frac{I_t}{I_0} dv}{PS(T)L} = \frac{A}{PS(T)L} \tag{2}$$

If the minimum detectable integrated absorbance $A_{min}$ is known, i.e., SNR (signal-to-noise ratio) = 1, the detection limit $X_{j,min}$ of species j can be expressed as:

$$X_{j,min} = \frac{A_{min}}{PS(T)L} \tag{3}$$

The temperature-dependent line intensity for a particular transition is determined by its line intensity $S$ at a reference temperature $T_0$ with the following equation:

$$S(T) = S(T_0) \left(\frac{Q(T_0)}{Q(T)}\right)_{T_0} \exp\left[-\frac{hcE_i''}{k}\left(\frac{1}{T} - \frac{1}{T_0}\right)\right] \cdot \left[\frac{1 - \exp(-hcv_{0,i}/kT)}{1 - \exp(-hcv_{0,i}/kT_0)}\right] \tag{4}$$

where $Q(T)$ is the partition function of the absorbing molecule ($CO_2$), $v_{0,i}$ is the frequency of the transition, $E''$ is the lower-state energy of the transition, $h$ is the Planck's constant ($6.626 \times 10^{-27}$ erg·s), $k$ is the Boltzmann's constant ($1.38 \times 10^{-16}$ erg/K), and $c$ is the speed of light in vacuum [35].

Temperature correction is required for TDLAS measurements in the environment with varying gas temperatures. According to Equation (2), in the case of the constant optical path and pressure, the ratio of integrated absorbance and line intensity, $A/S(T)$ is theoretically proportional to gas concentration $X$ in the same system. Therefore, the linear correction between $A/S(T)$ and gas concentration can be applied to compensate the errors caused by temperature influence as shown in the Equation (5):

$$y_{A/S(T)} = a + bx_X \tag{5}$$

In Equation (5), $a$ and $b$ are fitting coefficients. $a$ is the background signal without absorption, and $b$ is the gradient of the linear correction.

### 2.2. Wavelength Modulation Spectroscopy

Wavelength modulation spectroscopy could be used to improve the sensitivity of measurement. In WMS, the wavelength of the laser used is modulated by a combination of slow triangle wave and a fast-sinusoidal signal while the transmitted light is demodulated at a harmonic of the modulation frequency. For a sufficiently slow ramp, the frequency output of the laser can be expressed as [36]:

$$v(t) = \bar{v} + v_0 \cos(2\pi f t) \tag{6}$$

where $\bar{v}$ is the laser center frequency, $v_0$ is the modulation amplitude, and $f$ is the modulation frequency. The transmitted laser intensity can be expanded in a Fourier cosine series as:

$$I(t) = I_0(t)T[\bar{v} + v_0 \cos(2\pi f t)] = I_0(t) \sum_{k=0}^{k=\infty} H_k(\bar{v}, v_0) \cos(2\pi k f t) \tag{7}$$

where $H_k(\bar{v}, v_0)$ is the $k$th harmonic Fourier component of the transmission coefficient. The second harmonic Fourier coefficient which is given by:

$$H_2(x, m) = \frac{2XLPS(T)}{\pi\Delta v_c} \left\{ \frac{2}{m^2} \left[ \frac{2 + m^2}{(1 + m^2)^{1/2}} - 2 \right] \right\} \tag{8}$$

Two dimensionless numbers $x = 2(\bar{v} - v_0)/\Delta v_c$, $m = 2a/(\Delta v_c)$, $\Delta v_c$ (cm$^{-1}$) is the collisional linewidth (half width at half maximum, HWHM) of the probed transition, respectively; $m$ is the modulation index, a dimensionless parameter defined as the ratio of the wavelength-modulation amplitude to the linewidth $\Delta v_c$. CO$_2$ concentration $X$ is inferred explicitly from [37]:

$$X \propto \frac{I_0 P_{2f} \pi \Delta v_c}{S(T)PL} \left\{ \frac{2}{m^2} \left[ \frac{2+m^2}{(1+m^2)^{1/2}} - 2 \right] \right\}^{-1} \tag{9}$$

where $P_{2f}$ is the measured peak 2f signal at the line center $v_0$; therefore, the measured column density depends on the line intensity and linewidth of the transition, both of which vary with temperature.

For the optical thin condition due to the weak absorption in the near-infrared and low-gas concentration in industrial emissions, the peak 2f signal is linearly proportional to concentration:

$$y_X = a + bx_{P_{2f}} \tag{10}$$

The variation of line intensity and collisional linewidth caused by temperature change introduces errors to the above linear fitting. According to Equation (6), parameters like line intensity $S(T)$ and collisional linewidth $\Delta v_c$ are considered in the correction of temperature influence, which is given by:

$$y_X = a + bx_{\frac{P_{2f}\Delta v_c}{S(T)}} \tag{11}$$

## 3. Experiment Details

The exhaust of a power plant has a complicated composition, possibly including H$_2$O, SO$_2$, N$_2$, O$_2$, CO, and CO$_2$. According to the HITRAN2012 database [38], O$_2$ and SO$_2$ have no transitions near 1579 nm, and the strongest transition of H$_2$O is up to $10^{-26}$ in magnitude. Figure 1 shows the simulated absorbance of H$_2$O (0.8%), CO (0.001%), and CO$_2$ (10%) in the near-infrared at a typical condition of the power plant exhaust from 1578 to 1580 nm (6330–6335.5 cm$^{-1}$). The CO$_2$ absorbance near 1579.12 nm (6332.7 cm$^{-1}$) selected in this study is approximately 1000 times stronger than the transitions of other gases and isolated from CO interference transitions, while it is still accessible by distributed feedback diode lasers. Thus, this transition is sufficiently strong and ideal for in situ CO$_2$ measurement in the outlet ducts.

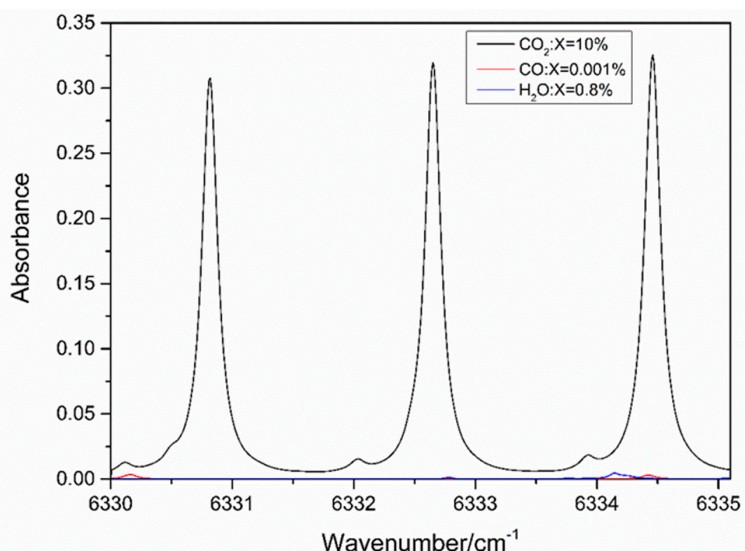

**Figure 1.** Simulated absorbance of CO$_2$ (10%), CO (0.001%), and H$_2$O (0.8%) from 6330–6335.5 cm$^{-1}$ based on the HITRAN (High Resolution Transmission) database (298 K, 1 atm).

The experimental setup for $CO_2$ concentration measurement is schematically shown in Figure 2. The setup consists of two main parts: gas preparation and laser detection. For the gas preparation, the $CO_2$ mixtures were generated by mixing the pure $N_2$ and $CO_2$ (0%–20%, interval of 2%) continuous gas flows controlled by two mass flow controllers (Seven Stars, Beijing, China). A gas mixer was used to obtain the uniform gas mixture. For the laser detection, the 1.58 μm DFB laser with ~5 mW output power was used as the laser source to exploit the $CO_2$ absorption line. A photodetector (SM05PD5A InGaAs Detector, THORLABS) was used to detect the transmitted laser beam through the test gas. Laser intensity and wavelength were varied by a combination of temperature and injection current using a commercial controller (PCI-1DA) with the real time spectrum display and capture features. The electrical signals received by the controller were processed into real-time spectral images. The wavelength was scanned (laser temperature at 35 °C, injection current scan between 30 and 120 mA) with a linear ramp of current from a function generator. The temperature and current tuning coefficients were measured to be 0.413 $cm^{-1}$/K and 0.049 $cm^{-1}$/mA, respectively. For the second harmonic signals, the laser controller contains lock-in amplification and demodulator for 2f signal extraction.

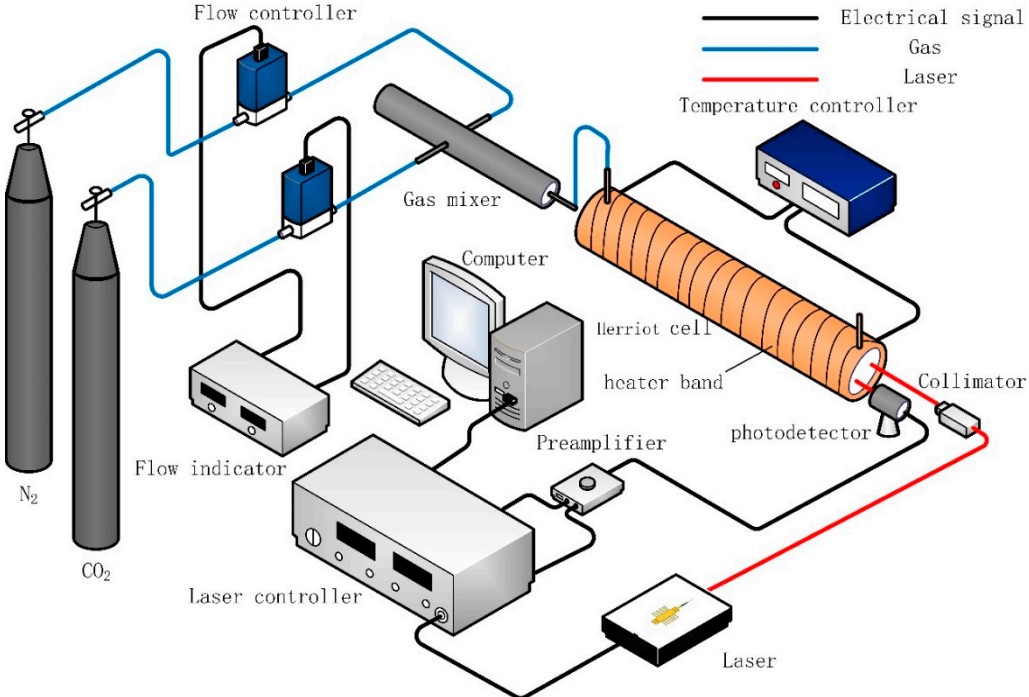

**Figure 2.** Schematic diagram of the experimental setup.

The laser beam was directed into a Herriot cell (29.5 cm long, 571 mL volume, effective optical path length of 20 m) by a collimator and detected by the InGaAs detector. The 20-meter optical path allowed the system to have sufficient detection sensitivity. The cell was mounted with two wedged (1.5°) sapphire windows to avoid unwanted interference fringes. Three type-K thermocouples were equally spaced along the center section of the Herriot cell to determine the temperature of $CO_2$; the maximum temperature difference observed among these thermocouples was <0.1 K from 298 K to 338 K. The gas pressure was continuously measured by a pressure gauge (YN60, full range 0.6 MPa) with an accuracy grade of ±0.015 MPa.

Before each experiment, the Herriot cell was purged with pure $N_2$ for 5 minutes to remove the residual gas. The cell was operated at 298–338 K (interval of 10 K) at an operational pressure of 1 atm. When a stabilized temperature was obtained, the test gases at six different $CO_2$ concentrations (10%–20%, interval of 2%) were measured with a single sweep of the laser. The continuous tuning range (at constant temperature) was within 5 $cm^{-1}$. The diode laser was scanned at a 5 Hz repetition

rate continuously, and the data were sampled at a frequency of 24 kHz to provide 4800 data points/sweep. The response time of real-time online monitoring can be further reduced by decreasing the sampling frequency. Each measured spectrum was obtained with 20 averaged sweeps, yielding a 4 s measurement time that was fast enough to satisfy the industrial application. Experiments using both direct absorption and wavelength modulation were repeated for three times to evaluate the repeatability of the sensor system.

## 4. Direct Absorption Measurement and Correction

In this part, the direct absorption spectra with and without temperature corrections are compared to analyze the measurement accuracy by relative errors. The *x*-axis of measured spectral data were sampling points, which were time domain signals. The frequency calibration of the spectrum was required to convert the time domain to the frequency domain. The simulations based on HITRAN2012 were used to calibrate the frequency domain. As shown in Figure 3, three wavenumbers in HITRAN2012 correspond to three points in measured spectral. Therefore, the curvilinear equation of quadratic polynomial was obtained to convert the time domain to the frequency domain. Figure 4 plots the measured incident signal $I_0$ and the direct absorption signals $I_t$ for 10% $CO_2$ at 318 K. The incident laser intensity $I_0$ was inferred by third-order polynomial fit in the far-wing parts without absorption. The Lorentz profile was chosen to fit the line shape for improving accuracy due to the domination of Lorentzian component at 1 atm. After fitting, the logarithmic ln $(I_t/I_0)$ was integrated over the whole frequency domain to calculate the integrated absorbance A. Thus, the concentration *X* can be calculated from Equation (2), with line intensity $S(T)$ obtained from Equation (4), total pressure *P*, and path length *L* as known parameters.

Figure 5 shows the measured absorption spectra of 16% $CO_2$ in the 6331.8–6333.6 cm$^{-1}$ spectral region at five different temperatures from 298 to 338K. The absorption signals in Figure 5 decrease with the increased temperature. Figure 6 plots the $CO_2$ line shape near 6332.7 cm$^{-1}$ (P = 1atm, T = 298 K, 10% $CO_2$ in $N_2$) well overlaid with the best-fit Lorentz profile. The obtained Lorentzian width of 0.15412 ± 0.0042 cm$^{-1}$ was broadened by pressure and temperature. The residuals of the curve to data is shown in the bottom panel of Figure 6, showing that the fits are similar. Table 1 shows the relative errors of direct absorption from 298 to 338 K. The error of inferred concentration obtained from the spectroscopic measurement generally rises with the increase of temperature. At 308 K, the biggest relative error was up to 5.66%, which cannot meet the requirement of industrial application.

To further explore the cause of measurement errors, line intensities from the HITRAN database were compared with measured line intensities which are calculated by Equation (4). With a certain $S(T)$, the obtained integrated absorbance *A* varies linearly with P·X·L where is the product of total pressure (*P* = 1 atm), absorption length (*L* = 2000 cm), and concentration (*X* = 10–20%, interval 2%). The relationship between *A* and P·X·L can be represented by a linear fitting and the slope of the fitting line is the intensity of the spectral line at the corresponding temperature from 298 K to 338 K, as depicted in Figure 7. Quality of the fit was very good in five temperatures, shown by R squared values from 0.99354 to 0.99875. As shown in Table 2, the calculated values show good agreement with the HITRAN values in variation tendency, but the former's gradient decreases more with the temperature rising. The discrepancy indicates that a 3–10% change of line intensity leads to 1–6% change in the measured $CO_2$ concentration.

Figure 8 depicts that both the integrated absorbance and the line intensity of 14% $CO_2$ decrease with the increase of temperature. The discrepancy leads to large errors in the concentration measurement, when the total pressure *P* and path length *L* are invariable. This information can be used to compensate these errors caused by change of temperature. According to the Equation (5), the least squares method is used for linear fitting between $A/S(T)$ and concentration *X* at different temperatures. Figure 9 shows the result of linear fit between $A/S(T)$ and concentration at T = 318 K. The fitting results show that the R squared value reaches 0.999, indicating a good linear relationship between $A/S(T)$ and concentration.

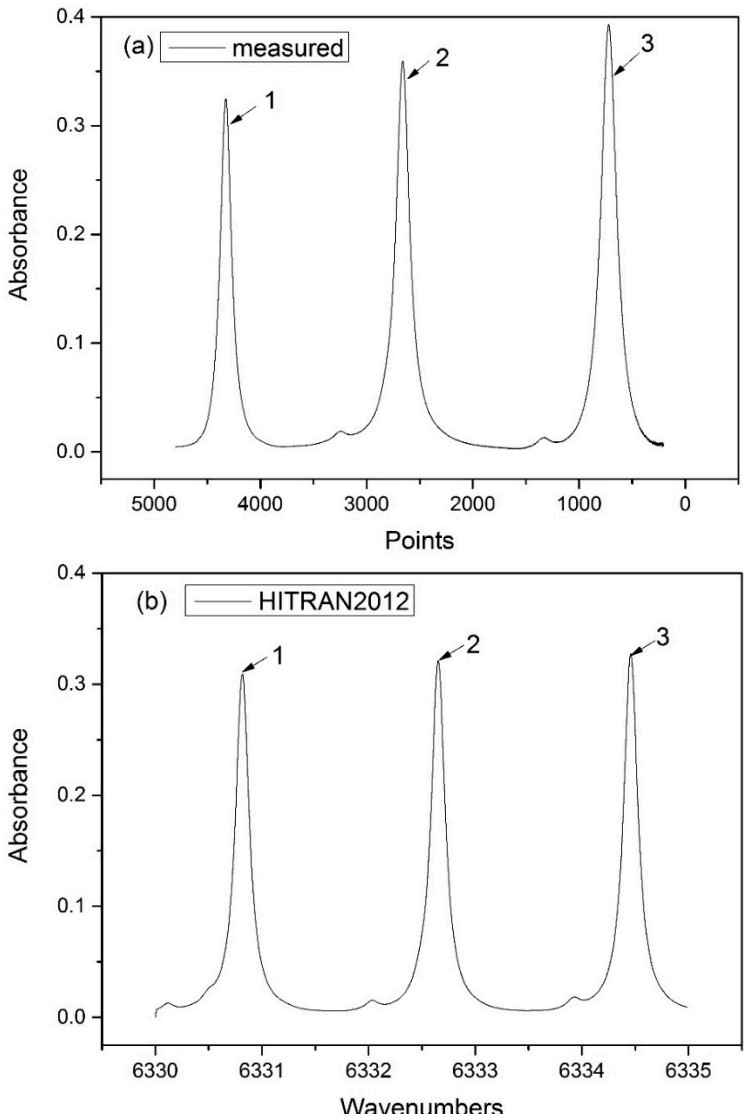

**Figure 3.** (**a**) Measured and (**b**) simulated absorbance of $CO_2$ at 296 K and 1 atm.

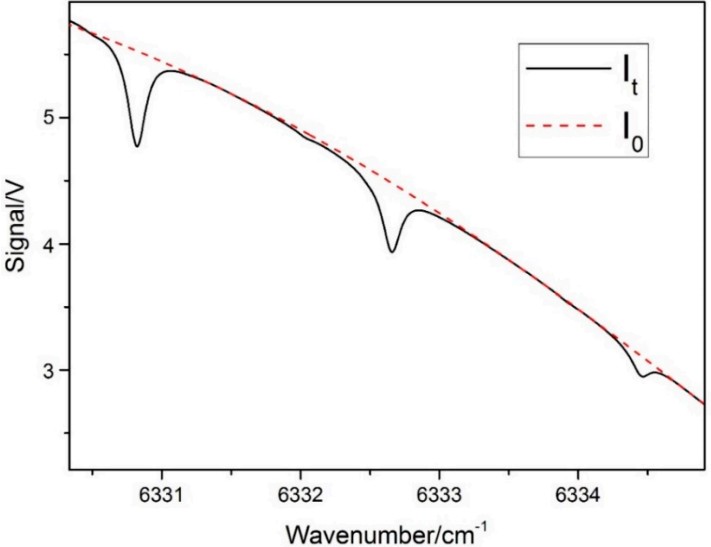

**Figure 4.** Direct absorption signals $I_t$ and baseline signal $I_0$ (T = 318 K, 10% $CO_2$, L = 20 m).

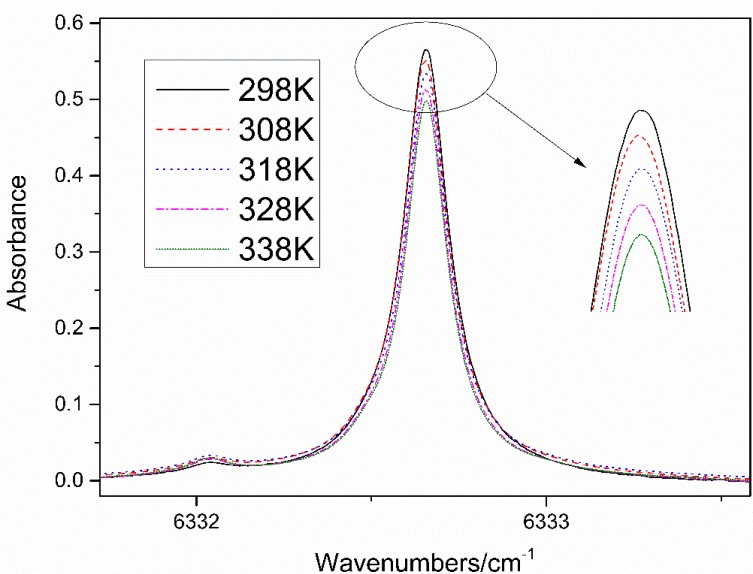

**Figure 5.** Absorption spectra of 16% $CO_2$ in the 6331.8 to 6333.6 $cm^{-1}$ spectral region at T = 298–338 K.

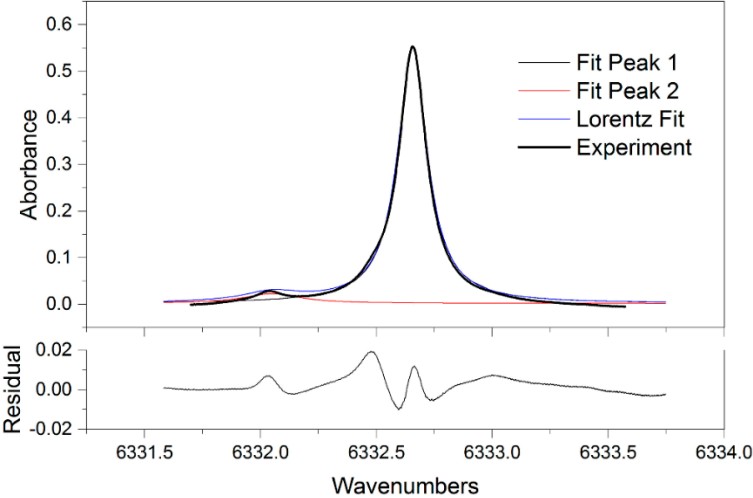

**Figure 6.** Measured $CO_2$ spectrum near 6332.7 $cm^{-1}$ with Lorentz fitting (P = 1 atm, T = 338 K, 18% $CO_2$).

**Table 1.** The relative errors of direct absorption results from 298 to 338 K.

| True Concentration | Inferred Concentration | | | | |
|:---:|:---:|:---:|:---:|:---:|:---:|
| | 298 K | 308 K | 318 K | 328 K | 338 K |
| 10% | 0.30% | 5.66% | 1.33% | 2.51% | −2.25% |
| 12% | −2.78% | −0.58% | 1.35% | −1.24% | −2.62% |
| 14% | 0.28% | −0.60% | −0.38% | −2.88% | −3.35% |
| 16% | −1.24% | 0.09% | 0.65% | −4.70% | −3.58% |
| 18% | 3.74% | 1.52% | 1.59% | 1.56% | −3.14% |
| 20% | −1.21% | 3.54% | 0.61% | −3.72% | −2.83% |

**Table 2.** Line intensity ($10^{-4}$ $cm^{-2} \cdot atm^{-1}$) values from database and calculated values.

| $S(T)$ | 298 K | 308 K | 318 K | 328 K | 338 K |
|:---:|:---:|:---:|:---:|:---:|:---:|
| Database | 3.8967 | 3.8147 | 3.7318 | 3.6483 | 3.5648 |
| Calculated | 4.0639 | 3.9329 | 3.8824 | 3.5011 | 3.2471 |

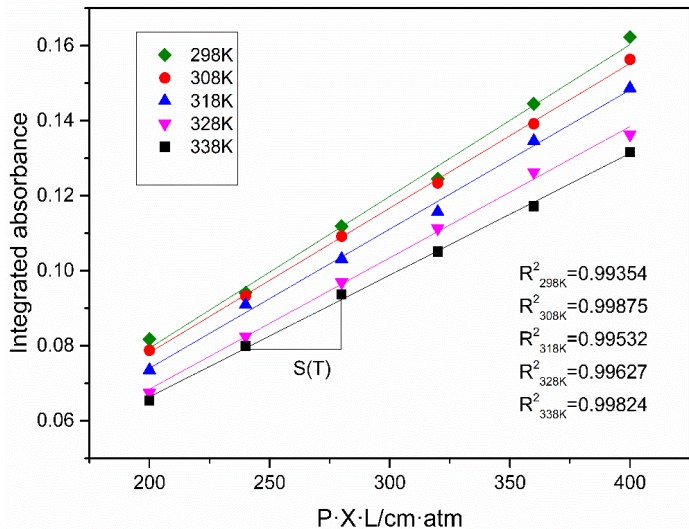

**Figure 7.** Integrated absorbance as a function of $P \cdot X \cdot L$ (T = 298–338 K, P = 1 atm).

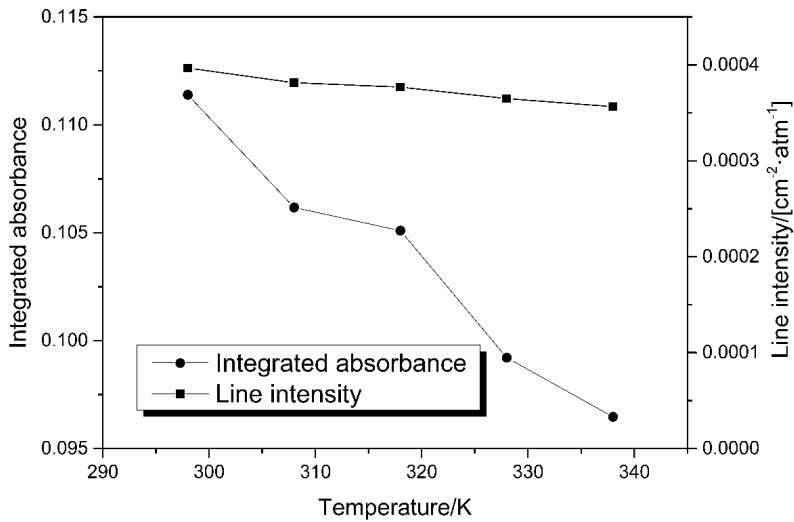

**Figure 8.** Integrated absorbance and spectral intensity with temperature change (14% $CO_2$).

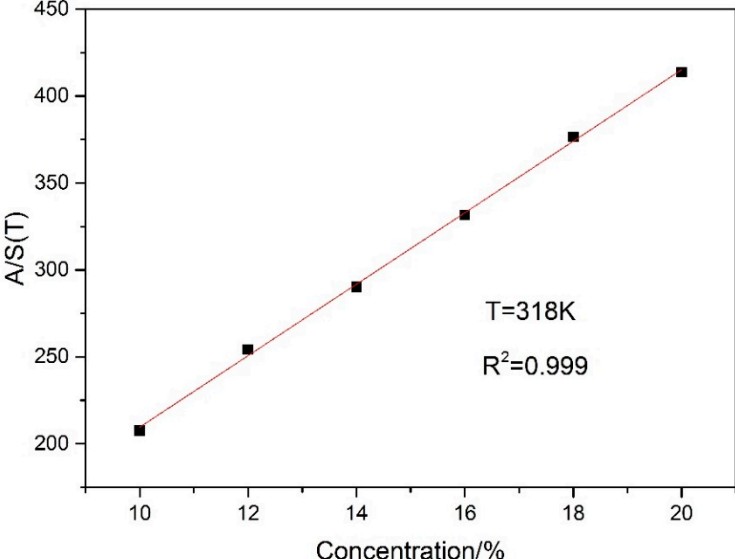

**Figure 9.** A linear fit between $A/S(T)$ and concentration at T = 318 K.

Figure 10 illustrates the absolute values of the relative error of the measured results before and after the correction. As shown in Figure 10, after temperature compensation, the maximum relative error is $-2.57\%$ corresponding to the measurement of 14% $CO_2$ at 318 K. However, the absolute error was only 0.36% and the relative error of the mean square value was less than 3.5%. All these measurements confirm that this temperature compensation enables the direct detection of $CO_2$ in outlet ducts temperature with a detection limit of $8.4 \times 10^{-5}$ at SNR = 1. Measurements of $CO_2$ concentration can be made for a variety of carbon emission sources, and the data are helpful for developing more accurate detection mechanisms.

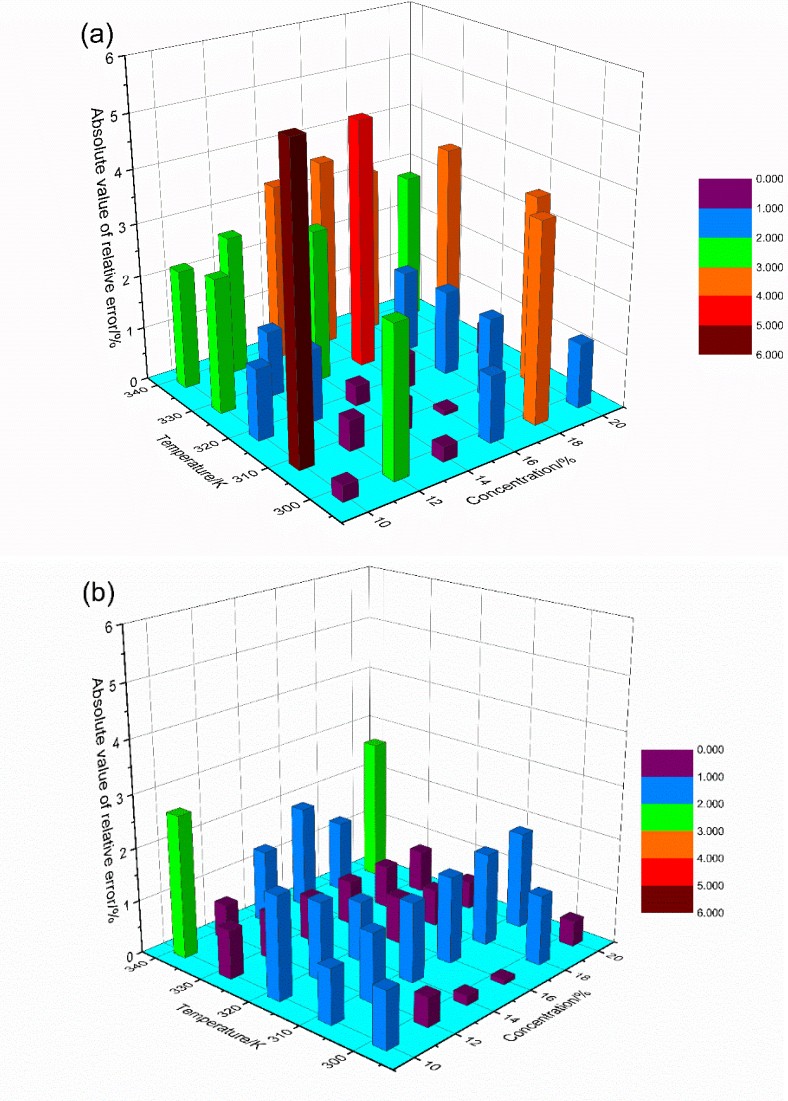

**Figure 10.** Absolute value of the relative error changes with concentration and temperature. (**a**) Before and (**b**) after compensation.

## 5. WMS-2f Measurement and Correction

Wavelength modulation spectroscopy (WMS) was applied to restrain a variety of background noise during measurement. Due to the low absorption of $CO_2$ in the near-infrared, the second harmonic (2f) signal was directly proportional to concentration [39]. When the optical path and pressure were constant, the WMS-2f peak amplitude was related to gas concentration and the wavelength modulation index $m$, which is dependent on the modulation current $I_{mod}$. Therefore, an appropriate choice of $I_{mod}$ was required to improve the detection precision.

The WMS-2f peak signals of $CO_2$ measured at different concentrations (10%–20%, interval 5%) and modulation currents (0 mA–18 mA, interval 1mA) are plotted in Figure 11. These measurements were conducted by controlling the gas flow controller and laser controller to maintain $CO_2$ concentrations and modulation currents. According to the experimental results shown in Figure 11, with the choice of $I_{mod}$ of ~15 mA, the WMS-2f peak signals are stable at all concentrations. Hence, $I_{mod}$ = 15 mA was selected in this study.

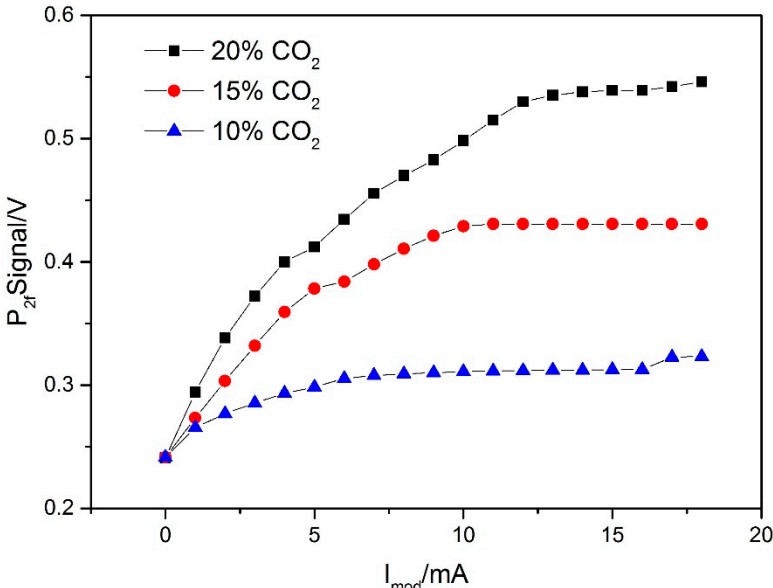

**Figure 11.** WMS (Wavelength modulation spectroscopy)-2f peak signals as a function of the laser modulation current $I_{mod}$ at different concentrations of $CO_2$.

In this study, the experimental operation was the same as the direct absorption experiment except a scanning frequency of 50 Hz and modulation frequency of 31.2 KHz were used. There was sufficient time to stabilize the second harmonic peak before the second harmonic data were recorded. Because the shape of the harmonic signal was affected by optical fringe noise and beam noise, the multi-sampling average (40 times) were used to eliminate the negative influences. Gas concentrations and peaks of second harmonic signals show good linear relationships at 298–338 K by employing the mean filtering process. The relationships can be expressed as the fitting of $X = a \cdot P_{2f} + b$, as Equation (10), which was obtained by the linear least squares method, and $P_{2f}$ was the peak of the second harmonic (V), and $X$ was gas concentration (%). The R squared value for linear fitting was >0.99 and the system reproducibility was tested after 60 min of continuous monitoring. In consideration of the temperature effect on line intensity and linewidth, the compensation method can be used to improve the accuracy of measurements. Following Equation (11), the linear relations between signals of $P_{2f} \cdot \Delta v_c / S(T)$ and corresponding concentrations were worked out by the least squares method. The averaged fitting R squared was improved up to 0.997, as shown in Figure 12. The peaks of WMS-2f signals at 6332.7 $cm^{-1}$ were plotted as a function of $CO_2$ concentration (linear fit equation: y = 0.4114 − 0.0167x, T = 318 K). The results are shown in Tables 3 and 4. Compared with the results without correction, the averaged relative error decreased by 0.24% and the total R squared value increased by 0.28%.

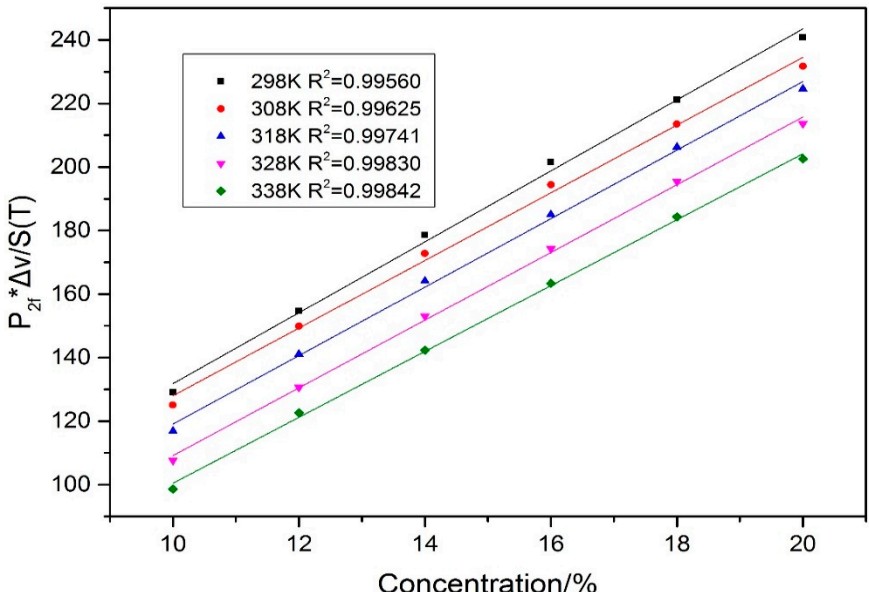

**Figure 12.** A linear fit of between $P_{2f} \cdot \Delta\nu_c / S(T)$ and concentration at different temperatures (298–338 K).

**Table 3.** The relative errors of wavelength modulation results before correction (298–338 K).

| True Concentration | Inferred Concentration | | | | |
|---|---|---|---|---|---|
| | 298 K | 308 K | 318 K | 328 K | 338 K |
| 10% | −2.86% | −3.02% | −2.60% | −2.62% | −2.67% |
| 12% | 1.49% | 0.99% | 0.43% | 0.46% | 1.04% |
| 14% | 0.68% | 1.64% | 1.84% | 1.73% | 1.31% |
| 16% | 2.89% | 1.43% | 1.15% | 1.51% | 1.06% |
| 18% | −1.50% | 0.31% | 0.54% | 0.10% | 0.32% |
| 20% | −0.89% | −1.64% | −1.65% | −1.48% | −1.35% |

**Table 4.** The relative errors of wavelength modulation results after correction (298–338 K).

| True Concentration | Inferred Concentration | | | | |
|---|---|---|---|---|---|
| | 298 K | 308 K | 318 K | 328 K | 338 K |
| 10% | −2.30% | −2.58% | −1.97% | −1.94% | −2.25% |
| 12% | 1.52% | 0.83% | 0.11% | 0.02% | 0.66% |
| 14% | −0.16% | 1.43% | 1.56% | 1.27% | 1.17% |
| 16% | 2.85% | 1.13% | 0.82% | 1.48% | 1.14% |
| 18% | −1.34% | 0.23% | 0.66% | 0.30% | 0.25% |
| 20% | −0.73% | −1.32% | −1.41% | −1.40% | −1.22% |

Figure 13 merges the inferred concentration values of all measurements at temperature 298–338 K and concentration 10–20%. The error bars represent the repeatability of three repeated experiments by SD. After compensation, the normal repeatability of $CO_2$ values of DA and WMS were 5.55% and 1.79% at a SD (1σ). The mean relative error, which was caused by the uncertainties of temperature, pressure, and absorption length measurements, decreased from 3.06% to 1.17%. For the wavelength modulation, the mean relative errors were 1.44% and 1.20%, before and after compensation respectively. The relative standard deviation (RSD) for DA and WMS were 0.42% and 0.31%, while the maximum relative errors and detection limits (SNR = 1) for DA and WMS were 2.7% and 2.85%, 0.0084% and 0.0017%. It should be noted that the experiment in this study examined only high $CO_2$ concentrations of 10–20%. As can be seen from the data, the effect on the direct absorption method was more outstanding.

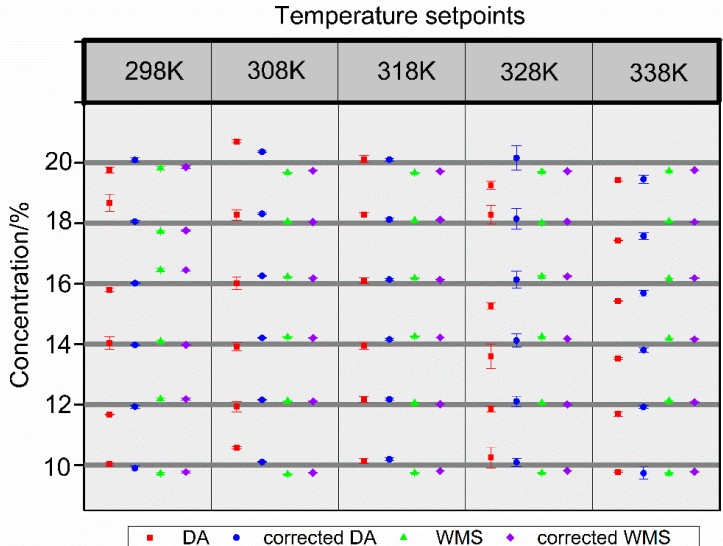

**Figure 13.** Measured concentration of all four measurements at temperatures 298–338 K and concentrations 10–20%.

## 6. Summary and Conclusions

Temperature compensations of $CO_2$ concentration measurements using TDLAS were reported. Absorption spectra at different temperatures were derived from HITRAN to find suitable transitions for in situ $CO_2$ measurements and recorded using DA and WMS. The temperature compensations based on temperature influence of line intensity were then applied to reduce measurement errors. Our experimental results show that there was an obvious decrease in the influence of temperature on signals after temperature compensation, and the detection limits of both methods meet the needs of the in situ $CO_2$ measurement. However, compared with WMS, DA is less sensitive to fluctuations of temperature and pressure, resulting in a superior long-term stability. The DA measurement based on temperature compensation has enough accuracy to be used for reliable determination of high concentration $CO_2$ in power plant exhausts.

**Author Contributions:** Conceptualization, X.Z. and S.Y.; Methodology, X.Z. and Z.L. (Zhenghui Li); Investigation, X.Z.; Writing—Original Draft Preparation, X.Z.; Writing—Review & Editing, W.R. and Z.L. (Zhimin Lu); Funding Acquisition, S.Y.

**Funding:** This research was funded by the Guangdong Province Train High-Level Personnel Special Support Program, grant number 2014TQ01N334, and the Guangdong Province Key Laboratory of Efficient and Clean Energy Utilization, grant number 2013A061401005.

**Acknowledgments:** The authors would like to thank the Guangdong Province Train High-Level Personnel Special Support Program (2014TQ01N334) and the Guangdong Province Key Laboratory of Efficient and Clean Energy Utilization (2013A061401005), South China University of Technology.

**Conflicts of Interest:** The authors declare no conflict of interest.

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
