# Peer review of "TDLAS Monitoring of Carbon Dioxide with Temperature Compensation in Power Plant Exhausts"

_applsci, doi:10.3390/app9030442_

Round 1

Reviewer 1 Report

Thank you for submitting your manuscript. It could use some proof reading and editing. The paper is generally well organized and presented. The introduction to the wavelength modulation technique is somewhat brief. It might be helpful to readers to expand a little.

1. Lines 41-42: “Error! Reference source not found” This is likely no intended?

2. Lines 44-45: “Doppler-broadening at high temperatures alters a larger portion of the integrated absorbance for a particular transition to the far-wing region.” Please elaborate the meaning especially with “Doppler-broadening” and “far-wing region”.

3. What is the typical exhaust temperature? Were the tests performed in those temperature ranges?

4. Depending on color for can be hard to read (Fig. 1).

5. Lines 174-176: It would be helpful to elaborate: “As shown in Figure 3, three wavenumbers in HITRAN2012 correspond to three points in measured spectral. Therefore, the curvilinear equation of quadratic polynomial is obtained to convert the time domain to the frequency domain.”

6. Line 207: “P·X·L where is the product of” considers “P·X·L where is the product of”.

7. Line 213: “but the former decreases more gradient with the temperature rising” not very clear. Is “gradient” supposed to be “gradually”?

8. Line 230: “As the shown in Fig.10” considers “As shown in Fig.10”.

9. It might be helpful to readers if the numbers of tests are provided with the statistics such as errors.

10. Line 285: “5.55 % and 1.79 % at 1σ” it might be helpful to readers to elaborate.

11. Please review Author Contributions and Funding sections.

Author Response

Dear reviewer,   We have studied the valuable comments from you carefully, and tried our best to revise the manuscript. The point-to-point responds to the reviewer’s comments are listed as following document.

Reviewer 2 Report

The paper investigates the impact of temperature changes on the absorption lines of CO2 and its impact on the accuracy of the concentration determination. This is motivated in the introduction, and references to other work is provided. The setup is described accurately, and each finding is backed up with data or figure. However, the manuscript is still unfinished and requires some cosmetic changes, which I will go through in the following:

Author correspondence

Keywords

Formatting of CO2 in abstract, and title if possible

Scaling of the figures; compare figure 3, 4, and 5 regarding font size and resolution

Author contributions

Acknowledgments -> Funding

Conflicts of Interest

In addition to cosmetic changes, I suggest other improvements to the manuscript.

L32: composition -> composite

L35-39: All the presented measurements are non-intrusive, yet this is listed as an advantage of TDLAS over others.

L41: references not found

L43: Tunable semiconductors are stated to be the nearly ideal source; this statement should be better explained by what ideal constitutes. Compact packaging and low cost can be achieved with LEDs as well.

L74+75: It is an oddly specific request to find spectroscopic work that matches application, environment, wavelength and gas all at the same time. It is not clear to me, why this exact parameter space is required. However, even without that sentence, the work has been motivated.

Introduction in general: Of the parameters that affect the spectroscopic measurements the most, temperature and concentrations are the most varying. Here, the authors chose to assume a fixed temperature measuring the concentration. Similar work exists when the concentration is fixed, but the temperature is measured (e.g. ref1, ref2). Linking both approaches would allow to benchmark the measurements against each other and should be possible with the existing data set.

L92: specie -> species

L105: An explanation of a physical interpretation of a and b would beneficial. It seems as a corresponds to absorption/signal at 0 concentration.

Figure 1: In Eq (2) at L90, Absorbance was introduced as an integrated quantity and should have no wavelength dependence. In Figure 1 the same quantity is used to describe the absorption spectrum, which is wavelength dependent. CO and H2O are difficult to find in this plot; if that is intentional the caption should explain why these concentrations have been assumed.

L139: internal to 2%; I assume that means interval of 2%

L162: same as above

Experimental details in general: Did you measure with continuous flow? Or closed cell after it has been filled?

Figure 3: Expand caption.

L196: I could not find the definition of inversion concentration. I assume this refers to the concentration obtained from the spectroscopic measurement. In the later tables inversion concentration is used to describe the (relative?) error of said concentration. The manuscript could benefit from a clearer statement, also regarding how those errors were calculated.

Figure 5: Absorbance on axis, absorption spectra in caption. Is the absorption spectrum a logarithmic quantity like the absorbance, or is it one a linear scale?

Figure 6: Most lines cannot be seen in print.

L230: "As the shown in Fig.10..."

Figure 10: At this resolution, the axis title are very difficult to read.

Figure 12: caption

Table 3: caption

Table 4: caption

---------

ref1: doi: 10.1007/s00340-016-6349-4

ref2: doi: 10.1063/1.4984252

In summary, this is a sound manuscript, with clear methodology and of interest for researchers in that field. After minor revisions, mostly of cosmetic nature, I would clearly suggest acceptance of the manuscript.

Author Response

(The authors gave the same response as above.)
